# Mediastinal Nodular Lesions Synchronous to Lung Carcinoma on Frozen Section: Trap and Lesson

**DOI:** 10.3390/diagnostics10110893

**Published:** 2020-10-31

**Authors:** Paola Parente, Marco Taurchini, Marina Castelvetere, Concetta Martina Di Micco, Domenico Greco, Paolo Graziano

**Affiliations:** 1Pathology Unit, Fondazione IRCCS Ospedale Casa Sollievo della Sofferenza, 71013 San Giovanni Rotondo (FG), Italy; m.castelvetere@operapadrepio.it (M.C.); p.graziano@operapadrepio.it (P.G.); 2Thoracic Surgical Unit, Fondazione IRCCS Ospedale Casa Sollievo della Sofferenza, 71013 San Giovanni Rotondo (FG), Italy; m.taurchini@operapadrepio.it (M.T.); mi.greco@live.it (D.G.); 3Oncology Unit, Fondazione IRCCS Ospedale Casa Sollievo della Sofferenza, 71013 San Giovanni Rotondo (FG), Italy; doctor.dimicco@email.it

**Keywords:** Thymoma, multiple thymoma, lung adenocarcinoma, signet ring cells

## Abstract

Thymoma is the most frequent neoplasm arising in the anterior mediastum. It usually presents as an enlarged central mass. In the literature, multiple thymoma is described as an unusual finding; rare variants have also been described, like the signet ring-like cell variant. Evidence of co-existence of signet ring-like cells and lymphocytes in small biopsies from nodular mediastinal lesions can lead to a diagnosis of metastatic carcinoma, mostly at frozen sections. Thymoma and pulmonary carcinoma are very rarely associated neoplasms. We present a case of two mediastinal lesions discovered during pulmonary carcinoma staging. At frozen section, a diagnosis of ‘epithelioid proliferation associated to lymphoid tissue’ was advanced on a sample of nodular lesions and of ‘carcinoma’ on pulmonary biopsy. Double AB Type Thymoma with a signet ring cell-like component, synchronous to pulmonary adenocarcinoma, was the diagnosis made on formalin fixed-paraffin embedded samples. Reporting the coexistence of these two entities can help pathologists and surgeons to establish the best management of similar patients.

## 1. Introduction

Thymoma is the most common tumor of the anterior mediastinum arising from the epithelial cells of the thymus [1]. Although thymoma is usually an indolent asymptomatic neoplasm, incidentally discovered on imaging survey, about 20% of patients with thymoma present with myasthenia gravis or other mass-related symptoms [1]. At computed tomography (CT), thymoma appears as a solitary well-defined mass located in the anterior mediastinum. Multiple thymoma is a very rare finding, consisting of multiple lesions located in the thymus with the same histological subtypes [2]. Unusual variants, such as signet ring-like cells, have also been described [3].

Adenocarcinoma is the most common histologic subtype of lung cancer, accounting for almost half of all pulmonary neoplasms [4]. The Word Health Organization (WHO) Classification of Tumors of the Lung, Pleura, Thymus and Heart, 2015 edition, classifies pulmonary adenocarcinoma according to its histologic growth pattern: lepidic, acinar, papillary, micropapillary, and solid [5]. The solid pattern shows a major component of tumor cells forming sheets that lack a recognizable feature of adenocarcinoma; in this setting, signet-ring features can also frequently occur. At an early clinical stage (I-II), surgery is the first treatment for lung cancer [6]. Without pre-operative diagnosis, frozen section is mandatory to establish a possible malignancy of the pulmonary lesion and to exclude an extra-regional lymph node metastasis [6].

Synchronous thymic malignancy and lung cancer is a very unusual finding, rarely described in the literature [7].

Here, we report the case of two mediastinal nodules incidentally found in a patient presenting with a pulmonary mass, without other systemic symptoms. The two nodules were both characterized on frozen sections as ‘epithelioid proliferation associated to lymphoid tissue’ and the pulmonary mass as ‘poorly differentiated carcinoma’, respectively.

## 2. Case Presentation

A 68 year-old man was admitted to our Hospital for cough and chest pain, without any previous history of cancer nor other systemic symptoms. Contrast-enhanced computed tomography (CT) of the chest showed a left pulmonary lesion with spiculated margins, 8 cm in diameter, partially solid with a cystic and necrotic component, involving the superior and inferior lobe and infiltrating both the pulmonary artery and the pulmonary fissure between the upper and lower lobe (Figure 1A,B). 

Regional lymph nodes were not detectable. In the antero-superior mediastinal region, inside the thymic compartment, two well circumscribed and encapsulated lesions were documented, both of 2.5 cm size (Figure 1A,B), radiologically referred to as enlarged lymph nodes of 3A and 4L station, respectively. The Positron Emission Tomography-CT scanner showed a hyper-captation of the pulmonary mass (SUV 21), confirming malignancy; mediastinal lesions showed no captation (SUV 0) (Figure 1C). We were unable to obtain a preoperative diagnosis. However, the presence on contrast-enhanced CT scan of the mediastinal lesions raised the suspicion of metastasis in the extra-regional lymph nodes (N2). Therefore, intraoperative characterization was programmed in order to confirm the suspicion for pulmonary neoplasm and to exclude the double extra-regional nodal metastasis in the mediastinal region, finalized to perform a left pneumectomy and regional lymphadenectomy with curative intent. Macroscopically, both mediastinal lesions showed lobular margins and a homogeneous and pink aspect on the cut section. The frozen section revealed monomorphic proliferation of round, epitheliomorphic cells admixed with numerous small lymphocytes, without necrosis and mitosis (Figure 2A,B).

An intraoperative diagnosis of ‘epithelioid proliferation associated to lymphoid tissue without localization of carcinoma’ was established. Smear from the biopsy of pulmonary mass showed a hypercellular sample with numerous epitheliomorphic cells, isolated and in micropapillary aggregates, with severe cyto-cariological atypia, necrosis and mitoses (Figure 2C). An intraoperative diagnosis of poorly differentiated carcinoma was established. Consequently, left pneumectomy with regional lymphadenectomy was performed. The histological evaluation of the formalin fixed-paraffin embedded (FFPE) section of the first nodular mediastinal lesion evidenced a lymphoepithelial proliferation with a microcystic growth pattern (Figure 3A), consisting of epithelial round monomorphic pancytokeratin (clone AE1-3) and p63 positive cells, lacking nuclear atypia, admixed with areas rich in small TdT and CD3 positive lymphocytes (Figure 4A,C).

FFPE sections from the second nodular mediastinal lesion showed overlapping findings with the first mediastinal lesion. A signet ring-like cells component was also observed (Figure 3B,C). Immunohistochemistry showed a positivity for pancytokeratin (clone AE1-3) and p63 in signet ring-like cells admixed with areas rich in small TdT and CD3 positive lymphocytes (Figure 4B,D). On FFPE sections, the two mediastinal lesions were defined as double AB Type Thymoma showing respectively a microcystic pattern and a microcystic pattern with signet ring-like cells. On the other hand, the pulmonary mass resulted as a primary adenocarcinoma with acinar and solid patterns. Either pleura infiltration or lymph node metastases were not observed (Figure 3D). Vascular hematic invasion was documented with a final staging of pT4a(PL0) pN0 V1 L0 R0 according to the 2017 UICC classification. 

All information regarding human material was managed using anonymous numerical codes, and all samples were handled in compliance with the Helsinki Declaration (http://www.wma.net/en/30publications/10policies/b3/). This was a retrospective study, and no extra tests were conducted on the patients’ material. We obtained, before drafting this paper, a written informed consent from the patient.

## 3. Discussion

Thymoma is the most common tumor of the anterior mediastinum arising from the epithelial cells of the thymus [1]. Thymoma shows a wide age distribution with a mean of 50-60 years and no major sex predilection. It is usually an indolent neoplasm, often associated with autoimmune diseases, the most common of which is myasthenia gravis [1]. At CT scanning, thymoma appears as a solitary and well encapsulated anterior mediastinal nodular lesion; its size is some centimeters in diameter, growing adjacent to the surrounding vascular structures but not infiltrating them. The WHO 2015 classification describes two major thymoma histotypes depending on whether the neoplastic cells and their nuclei have a spindle or oval shape and are uniformly bland (Type A Thymoma), or whether the cells have a predominantly round or polygonal appearance (Type B Thymoma) [5]. Type B Thymoma is further subdivided depending on the extent of the lymphocytic component into three subtypes (B1, B2 and B3). The thymoma combining type A and B features is designated as AB Type Thymoma [5]. Rare types of thymoma are reported as micronodular thymoma with lymphoid stroma, metaplastic thymoma, microscopic thymoma, lipo-fibroadenoma and sclerosing thymoma [1]. Furthermore, rare growth patterns of thymoma as microcystic and unusual components, such as signet ring-like cell, have also been described [3]. Generally, thymoma appears as a single lesion; moreover, very few cases of synchronous multiple thymomas are reported in the literature, with the same or different histological type, arising in the thymic compartment, in a well demarcated and encapsulated mass of the anterior mediastinum [2].

Adenocarcinoma is the most frequent pulmonary neoplasm [4] characterized by glandular malignant structures, different degrees of differentiation and histological patterns [5,8]. The solid pattern lacks glandular structures and it is the variant most often associated with signet ring-like cells; it is the most aggressive histotype, generally presenting node metastases and an advanced stage at clinical presentation [4]. In organ-confined diseases without metastasis, surgical resection of primitive tumor with regional lymph node sampling is the recommended treatment [6]. If metastasis to the mediastinal lymph nodes is histologically documented, surgical treatment is not allowed, but chemoradiotherapy followed by durvalumab consolidation therapy is expected [6]. A correct clinical and pathological preoperative staging is, then, mandatory for the best management [6].

Thymoma and pulmonary adenocarcinoma are very rarely associated entities [7,9]. Patients with thymoma exhibit an increased risk of developing second primitive extrathymic malignancies, with a frequency ranging between 8% and 28% [9]. In a multi-center study of 302 patients with thymoma, only 7 cases (less than 2%) were associated with lung primary carcinoma; of these, only 2 cases were synchronous to pulmonary carcinoma [7]. In the last cases, thymoma was described as a single mediastinal mass. 

Here, we describe an unusual case of double mediastinal thymoma, suspected of extra-regional metastases on radiological imaging, discovered while establishing the clinical staging of the lung cancer, and with signet ring-like cells component on FFPE slides.

This finding has not just a speculative aspect: we were lucky not to find the signet ring-like cells on the frozen section of the mediastinal lesions, which had raised a radiological preoperative suspicion of extra-regional lymph nodes. Signet ring-like cells admixed with lymphocytes in the contest of synchronous lung carcinoma would have led us to a diagnosis of extra-regional lymph node metastasis from lung cancer in intraoperative time; as a consequence, the surgeon would have not completed the left pneumectomy. A preoperative diagnosis of extra-regional lymph node metastasis would have led the patient to have undergone a second surgical treatment, with more risks and complexity involved than the first surgical session both for the surgeon and the patient.

## 4. Conclusions

Knowing the possibility, even if rare, of co-existence of pulmonary adenocarcinoma and multiple thymoma, even with a signet ring-like cell component, can be useful for the clinician, the pathologist and the surgeon when trying to establish the best patient management option in similar cases, without potentially dangerous delays in providing the surgical treatment.

## Figures and Tables

**Figure 1 diagnostics-10-00893-f001:**
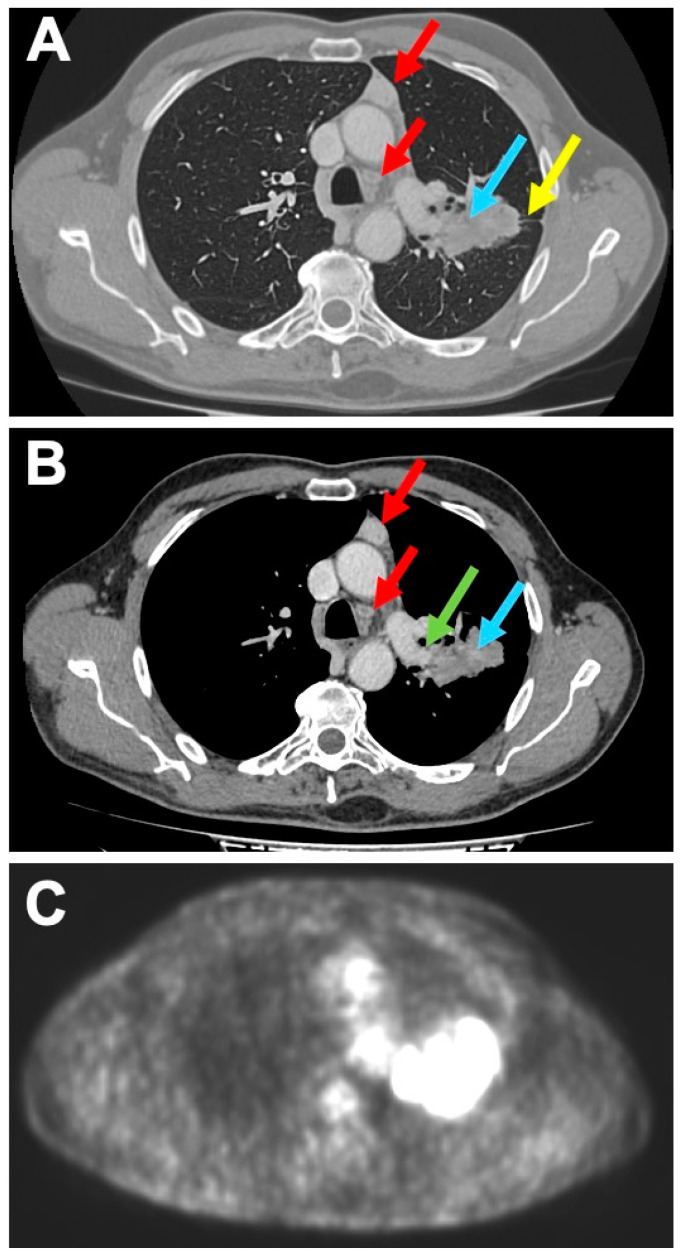
Contrast-enhanced computed tomography (CT) scan, lung window settings, showing two nodular mediastinal lesions (red arrows), a pulmonary mass (blue arrow), and a pulmonary fissure being infiltrated (yellow arrow) (**A**); contrast-enhanced CT scan, mediastinal window settings, showing two nodular mediastinal lesions (red arrows), a pulmonary mass (blue arrow), and the pulmonary artery being infiltrated (green arrow) (**B**); Positron emission tomography (PET) image showing pathological hyper-captation of the pulmonary mass without nodal mediastinal captation (**C**; axial level comprising the ascending and the distal tract of the aortic arch).

**Figure 2 diagnostics-10-00893-f002:**
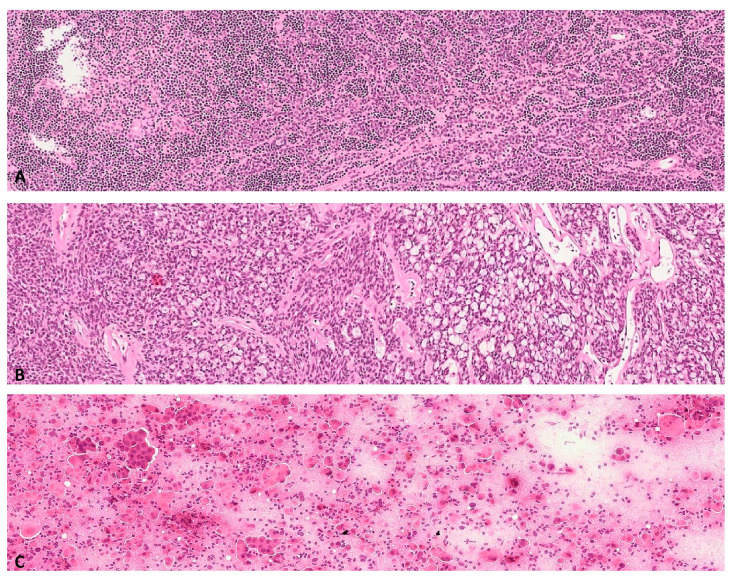
Epithelioid proliferation associated to lymphoid tissue on frozen sections from mediastinal lesions (**A**,**B**: Ematoxylin/eosin, 20×) and hypercellular smear with carcinomatous aggregates from the pulmonary mass (**C**: Ematoxylin/eosin, 30×).

**Figure 3 diagnostics-10-00893-f003:**
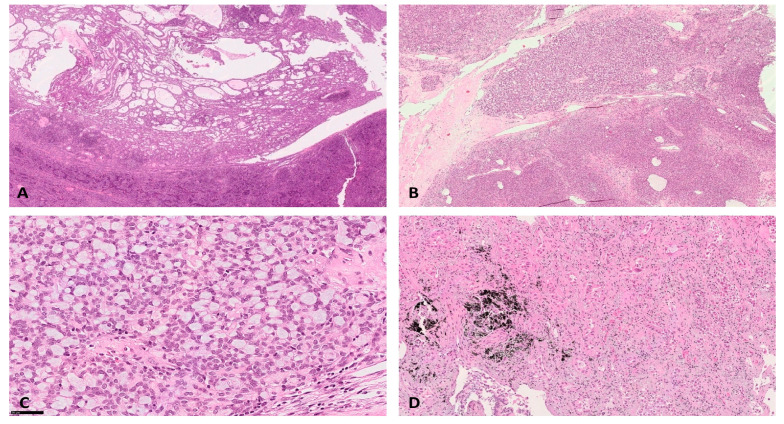
Formalin fixed-paraffin embedded (FFPE) sections showing microcystic AB Type Thymoma (**A**: Ematoxylin/eosin, 5×), microcystic AB Type Thymoma with signet ring-like cells features (**B**: Ematoxylin/eosin, 5×); signet ring-like cells in Thymoma (**C**, Ematoxylin/eosin, 40×) and pulmonary adenocarcinoma (**D**: Ematoxylin/eosin, 20×).

**Figure 4 diagnostics-10-00893-f004:**
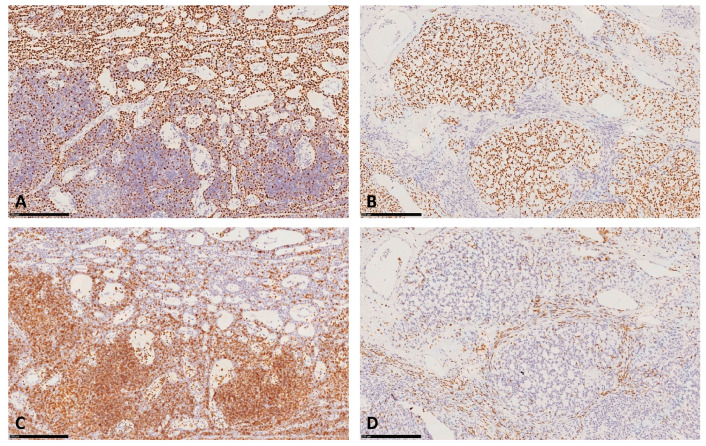
Immunostaining for p63 (15×) showing epithelial neoplastic cells positivity in microcystic AB Type Thymoma (**A**) and in microcystic AB Type Thymoma with signet ring-like cells features (**B**); Immunostaining for CD3 (15×) showing immature thymic lymphocytes positivity in microcystic AB Type Thymoma (**C**) and in microcystic AB Type Thymoma with signet ring-like cells features (**D**).

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
