# Peer review of "Mediastinal Nodular Lesions Synchronous to Lung Carcinoma on Frozen Section: Trap and Lesson"

_diagnostics, 2020, doi:10.3390/diagnostics10110893_

Round 1

Reviewer 1 Report

The article entitled “Mediastinic nodular lesions synchronous to lung carcinoma on frozen section: trap and teach” describe a case of lung adenocarcinoma accompanied by multiple thymoma lesions. It is interesting in that the co-existence of thymoma with lung cancer mimics the multiple station mediastinal lymph node metastases which hampers curative resection.

However, there are some concerns to be solved.

Major;

  • I agree with the authors' statement regarding the increased risk of second malignancy in cases with thymoma (page 5, line 127). However, the histological diagnosis with thymoma was made after surgical resection in the current case. Therefore, it is more important to state the strategy to discriminate mediastinal lymph node metastases from other nodules including thymoma in lung cancer patients with mediastinal nodules. Therefore, the statement on the diagnostic value of contrast-enhanced CT, MRI, and PET-CT would be more informative for the leaders.
  • It is difficult to make an accurate decision with one slice of CT image, however I wonder if the left upper lobectomy plus the left superior segmentectomy (S6) could have been better for the post-operative lung function. I want to describe the reason why pneumonectomy was inevitable.
  • The authors stated in the discussion that systemic therapy is expected in case of mediastinal lymph node metastases. I think chemoradiotherapy is the most appropriate strategy for stage III non-small cell lung cancer.

Minor;

  • It is difficult to qualitatively evaluate the mediastinal nodules via CT images with lung window settings. Therefore, the contrast-enhanced CT image with mediastinal window setting at the same axial level should be presented as well.
  • The evaluation of the difference in FDG accumulation between the lung primary site and mediastinal nodule is important to distinguish lymph node metastases from others, including non-specific lymph nodes and sarcomatoid reaction due to lung cancer. Thus, PET-CT image should be presented with the SUVmax values of the mediastinal nodules, though the authors described the mediastinal lesions were “not pathological” without referring to the SUVmax value.
  • The authors described the mediastinal nodules as 2.5 cm of size. However, these nodules should be smaller, considering the size of adjacent innominate veins and the diameter of the lung cancer (8 cm).
  • I prefer “staging” to “stadiation”.
  • I prefer “mediastinal” to “mediastinic”.
  • The expression “in a contest of synchronous lung cancer” (line 137) could be “in the context of synchronous lung cancer”.

Author Response

Response to Reviewer 1 Comments

Reviewer 1: The article entitled “Mediastinic nodular lesions synchronous to lung carcinoma on frozen section: trap and teach” describe a case of lung adenocarcinoma accompanied by multiple thymoma lesions. It is interesting in that the co-existence of thymoma with lung cancer mimics the multiple station mediastinal lymph node metastases which hampers curative resection. However, there are some concerns to be solved.

We are grateful to Reviewer 1 for his/her kind consideration of our paper and useful suggestions. The paper has been reviewed as follows:

Major:

  • I agree with the authors' statement regarding the increased risk of second malignancy in cases with thymoma (page 5, line 127). However, the histological diagnosis with thymoma was made after surgical resection in the current case. Therefore, it is more important to state the strategy to discriminate mediastinal lymph node metastases from other nodules including thymoma in lung cancer patients with mediastinal nodules. Therefore, the statement on the diagnostic value of contrast-enhanced CT, MRI, and PET-CT would be more informative for the leaders.

Contrasted-enhanced CT imaging of mediastinal nodes, in presence of pulmonary radiologically malignant mass, was suggestive of pathological (metastatic) extra-regional lymph nodes (we added the station 3A and 4L, respectively, at line 71-72, in red). Although mediastinal lesions were not ipercaptant on PET imaging, malignant nature of both mediastinal lesions could not be ruled out confidentially. We specified it on line 74-76. For this reason, an intraoperative histological characterization on frozen section of both lesions was mandatory.

  • It is difficult to make an accurate decision with one slice of CT image, however I wonder if the left upper lobectomy plus the left superior segmentectomy (S6) could have been better for the post-operative lung function. I want to describe the reason why pneumonectomy was inevitable.

We changed Figure 1 and we added one slice showing infiltration of both the pulmonary fissure (Fig1, A, yellow arrow) and the pulmonary artery (Fig1, B, yellow arrow). We added this information in the text (line 58-59). In order to obtain an oncologically correct resection of R0 margins, it was necessary to perform a pneumonectomy, because the neoplasm infiltrated both the pulmonary artery and the pulmonary fissure between the upper and lower lobe.

  • The authors stated in the discussion that systemic therapy is expected in case of mediastinal lymph node metastases. I think chemoradiotherapy is the most appropriate strategy for stage III non-small cell lung cancer.

We apologize for the incorrect statement: a more accurate description could be ‘neoadjuvant therapy’ (line 154, RED).

Minor:

  • It is difficult to qualitatively evaluate the mediastinal nodules via CT images with lung window settings. Therefore, the contrast-enhanced CT image with mediastinal window setting at the same axial level should be presented as well.

We changed Figure 1 and we added one slice showing CT image with the mediastinal window setting (Fig. 1, B) at the same axial level as CT image with lung window settings (Fig. 1, A).

  • The evaluation of the difference in FDG accumulation between the lung primary site and mediastinal nodule is important to distinguish lymph node metastases from others, including non-specific lymph nodes and sarcomatoid reaction due to lung cancer. Thus, PET-CT image should be presented with the SUVmax values of the mediastinal nodules, though the authors described the mediastinal lesions were “not pathological” without referring to the SUVmax value.

We specified the SUV of mediastinal nodes (SUV 0) in the text (line 73-74 in RED) and we added a PET image in the same axial level as CT scan showing hypercaptation of the lung mass and no captation of the mediastinic nodes (Figure 1, C).

  • The authors described the mediastinal nodules as 2.5 cm of size. However, these nodules should be smaller, considering the size of adjacent innominate veins and the diameter of the lung cancer (8 cm).

The maximum diameter of the mediastinal nodes and that of the lung cancer are described in the text. It was not possible to represent both maximum diameters in the same axial level.

  • I prefer “staging” to “stadiation”.

We changed ‘stadiation’ into ‘staging’.

  • I prefer “mediastinal” to “mediastinic”.

We changed ‘mediastinic’ into ‘mediastinal’.

  • The expression “in a contest of synchronous lung cancer” (line 137) could be “in the context of synchronous lung cancer”.

We changed ‘in a contest of synchronous lung cancer’ into ‘in the context of synchronous lung cancer’  (line 166 in RED)

We hope the required integrations may be deemed satisfactory and exhaustive.

Reviewer 2 Report

In the manuscript Parente et al. the authors report a case of double mediastinum lesion detected together with a lung adenocarcinoma.

I have the following comments:

1) the manuscript requires English editing.

2) in the materials and method section the authors should specify that their institutional review board approved the study and that written informed consent was obtained from the patient.

3) In figure 1 please add arrows to clearly indicate lesions and pulmonary mass. Also please clearly add the size of each one.

4) As a suggestion, I think would be easier for the reader if the panels in figure 2 had some space between them.

5) In the text were the authors refer to figure 3 they mention the presence of CD5, CD3 positive cells in both lesions. I think a higher magnification would be useful to clearly identify them.

Author Response

Response to Reviewer 2 Comments

Reviewer 2: In the manuscript Parente et al. the authors report a case of double mediastinum lesion detected together with a lung adenocarcinoma.

I have the following comments:

We are grateful to Review 2 for her/his kind considerations about our paper and for useful comments and suggestions. The paper has been reviewed as follows:

  • The manuscript requires English editing.

1) The final draft has been linguistically and stylistically revised by a mother-tongue translator and proofreader.

  • In the materials and method section the authors should specify that their institutional review board approved the study and that written informed consent was obtained from the patient.

2) We specified in the ‘Materials and method section’ that: ‘All information regarding human material was managed using anonymous numerical codes, and all samples were handled in compliance with the Helsinki Declaration (http://www.wma.net/en/30publications/10policies/b3/). This was a retrospective study, and no extra tests were conducted on the patients’ material. We obtained a written informed consent from the patient before writing this case report’ (line 123-127, in RED).

  • In figure 1 please add arrows to clearly indicate lesions and pulmonary mass. Also please clearly add the size of each one.

3) We changed Figure 1. Specifically, the red arrows indicate the mediastinal nodes; the blue arrow indicates the lung lesion; the yellow arrows indicate the pulmonary fissure (Fig1, A) and the pulmonary artery (Fig1, B).

The maximum diameter of the mediastinal nodes and that of the lung cancer are described in the text. It was not possible to represent both maximum diameters in the same axial level.

  • As a suggestion, I think would be easier for the reader if the panels in figure 2 had some space between them.

4) We added some space between panels in Figure 2 and in Figure 3.

5) In the text were the authors refer to figure 3 they mention the presence of CD5, CD3 positive cells in both lesions. I think a higher magnification would be useful to clearly identify them.

5) We apologize for the incorrect statement. Epithelial cells in the AB Thymoma are not CD5 positive. We have corrected the sentence (line 98, in red) and added Figure 4, which is representative of immunoistochemistry of both the p63 positive epithelial neoplastic component and the immature CD3 positive thymic lymphocytes.

We hope the required integrations may be deemed satisfactory and exhaustive.

Round 2

Reviewer 1 Report

The manuscript has been well revised.

However, I think the most appropriate treatment for the current case is chemoradiotherapy, if the mediastinal nodules were histologically diagnosed as metastases.

Since the submitted image of mediastinal window setting showed multiple station N2, I recommend chemoradiotherapy followed by durvalumab consolidation therapy in such cases.

Reviewer 2 Report

The authors have addressed all my comments.

Author Response

Response to Reviewer 2 Comments

Reviewer 2: The authors have addressed all my comments.

We are grateful to Review 2 for her/his previous appropriate considerations about our paper and for useful comments and suggestions and for her/his kind final approvation.